# Zebrafish Xenografts Unveil Sensitivity to Olaparib beyond BRCA Status

**DOI:** 10.3390/cancers12071769

**Published:** 2020-07-02

**Authors:** Ana Beatriz Varanda, Ana Martins-Logrado, Miguel Godinho Ferreira, Rita Fior

**Affiliations:** 1Champalimaud Centre for the Unknown, Champalimaud Foundation, 1400-038 Lisbon, Portugal; ana.varanda@research.fchampalimaud.org (A.B.V.); ana.logrado@neuro.fchampalimaud.org (A.M.-L.); miguel-godinho.ferreira@unice.fr (M.G.F.); 2Institute for Research on Cancer and Aging of Nice (IRCAN), Université Côte d’Azur, U1081 UMR7284 UNS, 06107 Nice, France

**Keywords:** breast cancer, BRCA1/2, ionizing radiation, olaparib, PARP, radiotherapy, xenografts, zebrafish

## Abstract

Poly (ADP-ribose) polymerase (PARP) inhibition in BRCA-mutated cells results in an incapacity to repair DNA damage, leading to cell death caused by synthetic lethality. Within the treatment options for advanced triple negative breast cancer, the PARP inhibitor olaparib is only given to patients with BRCA1/2 mutations. However, these patients may show resistance to this drug and BRCA1/2 wild-type tumors can show a striking sensitivity, making BRCA status a poor biomarker for treatment choice. Aiming to investigate if the zebrafish model can discriminate sensitivities to olaparib, we developed zebrafish xenografts with different BRCA status and measured tumor response to treatment, as well as its impact on angiogenesis and metastasis. When challenged with olaparib, xenografts revealed sensitivity phenotypes independent of BRCA. Moreover, its combination with ionizing radiation increased the cytotoxic effects, showing potential as a combinatorial regimen. In conclusion, we show that the zebrafish xenograft model may be used as a sensitivity profiling platform for olaparib in monotherapy or in combinatorial regimens. Hence, this model presents as a promising option for the future establishment of patient-derived xenografts for personalized medicine approaches beyond BRCA status.

## 1. Introduction

In the process of tumorigenesis, DNA damage recognition and repair can be arrested by a failure in one or several components of the genomic maintenance machinery, resulting in genomic instability [1]. From the vast array of lesions induced in DNA, double-strand breaks (DSBs) are the most deleterious. They can emerge in cells either in a programmed manner or in a random way, such as after exposure to agents like ionizing radiation (IR). Complete repair of DSBs, including restoration of molecular integrity and sequence, can be achieved with higher precision by homologous recombination repair (HRR) [2,3].

Cancer-associated genes, BRCA1/2, are known to play an important role in repair of DSBs via HRR, and their mutations can lead to gross chromosomal rearrangements and genome instability [4]. Thus, these proteins have been described to be involved in many different types of cancer, ranging from breast, ovarian and prostate cancer, to pancreatic, gallbladder and stomach cancers, or even melanoma [5]. To date, cancer treatment is prescribed according to wide-ranging guidelines, approved and selected after a demonstration of average efficacy and safety. According to the treatment guidelines for advanced triple negative breast cancer (TNBC), patients are only eligible for treatment with the poly(ADP-ribose) polymerase (PARP) inhibitor olaparib after undergoing first-line systemic non-targeted chemotherapy, and only if they carry a BRCA1/2 mutation [6].

PARPs are a family of enzymes that are involved in multiple cellular processes, ranging from cell replication to cell death. However, PARPs are mostly known for their involvement in DNA damage response (DDR) pathways, particularly in the response to single-strand DNA breaks (SSBs) as a component of the base excision repair (BER) complex [7]. Specifically, olaparib inhibits PARylation by competing with the binding of NAD+ to PARP1, PARP2, and PARP3. Additionally, olaparib traps PARP1 and PARP2 on DNA [8,9]. In this way, PARP inhibition impairs SSB repair, leading to genomic instability and cell cycle arrest. Furthermore, this leads to PARP trapping onto DNA with subsequent stalling of replication forks, contributing to the formation of DSBs. In normal conditions, a cell can repair these DSB by HRR and resume their life cycle. Yet, in BRCA1/2 mutants where HRR is impaired, inhibition of PARP leads to cell death. This illustrates the concept of synthetic lethality [10,11]. This concept has been the rationale for the original development of PARP inhibitors (PARPi), and it is also the reason why BRCA1/2 mutations are considered mandatory biomarkers for treatment with these therapies. However, many patients with BRCA1/2 mutations show resistance to the drug, whereas wild-type patients can be strikingly sensitive [12,13]. It is now recognized that many other genes involved in DDR pathways can be altered, meaning that BRCA status on its own is a poor biomarker for treatment choice. Thus, the challenge remains to discover a reliable test able to identify patients that are more likely to respond to PARPi therapy. To this end, several recent studies have focused on finding biomarkers capable of predicting the DDR deficient or BRCAness phenotype of patients [14]. A number of these assays test for mutations, promoter methylation or transcription status of large panels of genes related to DDR pathways, including BRCA1/2, 53BP1, RAD51B/C/D, XRCC1; ERCC1; XPF, USP15, PALB2, BARD1, BRIP1, ATM, FAAP20, CHEK2, FAN1, FANCE, FANCM, and POLQ [14]. Even in the TOPARP-B trial, a phase II randomized trial of PARPi olaparib for metastatic castration resistant prostate cancers (mCRPC) with DDR alterations, which used a panel of 133 genes, some tumors that are considered DDR-deficient do not respond to PARP inhibition, whereas others that lack DDR mutations do [15,16]. This lack of predictive capacity can be associated with the complexity and redundancy of DDR pathways or with the restoration of HRR function through various resistance mechanisms [14,16,17,18,19,20].

Other assays aim to analyze the “genomic scar” imprinted in the genome by the loss of HRR function—i.e., the mark left behind by DDR deficiencies—independently of the impaired member(s) of the pathways [14]. However, it has been described that mutations in several HRR-related genes may be reverted or suffer secondary mutations, restoring HRR function, thus once again leading to a lack of predictive capacity associated with this type of approach [21,22,23,24,25,26,27]. Moreover, these omics methods do not consider all the possible genetic interactions which may occur between different subclones or with the tumor microenvironment (TME). These are a “frozen picture” of dead cancer cells, which lack quantification of their response to direct perturbations.

Several studies are aiming to replace mutation-specific assays by functional assays that read the functional state of the DNA repair pathways. One example is the detection of RAD51 foci in the S/G2 cell phase of the cell cycle, which is indicative of HRR proficiency. Their absence suggests HRR deficiency and PARPi sensitivity, independently of which gene is affected. This promising assay is performed in routine formalin-fixed paraffin-embedded tumor samples prior to or after a DNA damaging insult [28,29]. However, it entails many technical challenges, such as fixation artifacts, reliable quantification of foci and timing of assessment [14].

Here, we tested a different approach. Instead of evaluating the functional state of HR and, thereafter, indirectly infer PARP sensitivity, we directly challenge tumor cells with olaparib and measure the resulting anti-tumoral effect in a quick in vivo model with single cell resolution—the zebrafish xenograft. Recently, we showed that zebrafish xenografts have resolution to reveal intra- and inter-tumor heterogeneity and differential response to chemo- and radio-therapy in just one week [30,31]. This model represents not only a feasible clinical platform for therapy screening for personalized treatments within the clinical decision timeframe, but also a valuable tool to study the interlink between tumor microenvironment and therapeutic response [32,33]. This is particularly important in PARPi therapies, since a plethora of different roles for PARP proteins have been described, from their role in DDR to functions related to the TME, such as angiogenesis, epithelial to mesenchymal transition (EMT) (involved in the metastatic process), or the interplay with the immune system [34,35,36]. Therefore, a model that can go beyond the cell’s intrinsic responses can be of great value.

In this work, we use zebrafish xenografts as a sensitivity profiling platform for olaparib not only in monotherapy, but also in combination with IR. When in combination with PARPi, IR can additively or synergistically enhance the accumulation of DNA damage, consequently activating cell-death signaling pathways [37]. Firstly, we generated zebrafish xenografts with the original VC8/VC8-B2 isogenic cell lines, differing only in BRCA2 status. In a second and more translational approach, the model was tested with a panel of human triple negative breast cancer (TNBC) cell lines with different BRCA status. Our results open the possibility of using the zebrafish larvae xenograft model as a PARPi screening platform for future personalized medicine or co-clinical trials design.

## 2. Results

### 2.1. Phenotypes Associated with Different BRCA2 Status Are Observed in Zebrafish Xenografts

As a proof of concept, we started by investigating the isogenic Chinese hamster cell lines VC8 and VC8-B2, which differ only in their BRCA2 status—VC8 is BRCA2 deficient whereas VC8-B2 was reconstituted with the human BRCA2 gene [38]. These were the original cell lines used in the groundbreaking work that identified PARPi as a synthetic lethality therapeutic strategy for BRCA-deficient tumors [39,40]. This isogenic model has the capacity to highlight the different effects of PARPi depending on the BRCA2 status.

To generate the xenografts, tumor cells were fluorescently labelled and injected into the perivitelline space (PVS) of 2 days post-fertilization (2dpf) zebrafish embryos (Figure 1A). Xenografts were characterized according to several hallmarks of cancer, such as cell proliferation, cell death, angiogenic potential and metastization. At 5 days post-injection (5dpi), we could not detect any significant difference between the two isogenic xenografts regarding cell proliferation, measured by mitotic index (Figure 1C,C’,G; *p* = 0.1509), which is in accordance with previous in vitro reports for these cell lines [38]. We could, however, detect other differential behaviors.

Quantification of Ki-67 revealed that BRCA2 mutated tumors display a significantly higher Ki-67 index (Figure 1D,D’,H, * *p* = 0.0496). Although traditionally viewed as a proliferative marker, it has been shown that Ki-67 operates as a chromatin organizer and does not always correlate with proliferative potential [41]. Furthermore, it has been reported to be highly expressed in BRCA2 mutated breast and ovarian cancers, in relation to sporadic WT tumors [42], similarly to what we found in VC8^BRCA2mut^ zebrafish xenografts (Figure 1D’,H).

Analysis of basal cell death, assessed by the expression of activated caspase 3, the last effector in the pathway of apoptosis, shows that it was also significantly higher in BRCA2 mutated tumors (Figure 1E,E’,I), once again similarly to what has previously been described in vitro [39].

Numerous studies report an association between BRCA1/2 mutations and an increased expression of molecules that correlate with a higher angiogenic potential [43,44]. To analyze tumor-induced angiogenesis, we generated xenografts in *Tg(fli1:eGFP)* zebrafish hosts, which have the vasculature labelled with enhanced Green Fluorescent Protein (eGFP) [45]. At 5dpi, we analyzed vessel recruitment by quantifying total vessel density (GFP quantification per tumor area), but no statistically significant differences were found (Figure 1F,F’,J).

Finally, we analyzed the ability to establish micrometastasis in the caudal hematopoietic tissue (CHT) region, the furthest place from injection and a hotspot for tumor cell colonization in zebrafish [31]. However, both tumor cell lines showed a very low metastatic potential, with VC8-B2 with ~4.8% of xenografts showing micrometastasis in the CHT (3/62) and only ~1.2% in VC8 (1/83 xenografts, VC8-B2 vs. VC8 Fisher test *p* = 0.31, from a pool of four independent experiments).

In summary, the VC8-B2/VC8 xenografts behave similarly, as expected for isogenic cell lines, with exception to its Ki-67 and apoptotic basal levels, which is in accordance with the BRCA2 role in DDR and the available literature [38,39].

### 2.2. VC8-B2 and VC8 Zebrafish Xenografts Show Different Sensitivities to Treatment with Olaparib Alone and in Combination with IR

In general, PARPi are not given as monotherapy. Instead, they are usually sequentially combined with DNA damaging agents like platinum-based chemotherapy [46,47,48] or IR, since these therapies have the potential to sensitize for each other [49,50,51,52,53]. Therefore, we tested olaparib in our model not only in monotherapy, but also in combination with IR, using a 25Gy protocol that we recently developed and validated [30].

We started by testing different concentrations of olaparib diluted in embryonic medium (E3), having as a reference the maximum plasma concentration (C_max_) reported in patients—5 μM [54]. Since olaparib treatment did not cause any mortality in zebrafish embryos, we selected 50 μM as our working concentration (Appendix A, see details in Section 4).

After protocol optimization, VC8-B2^BRCA2wt^ and VC8^BRCA2mut^ xenografts were generated and, at 24 hpi, xenografts were randomly distributed in the different conditions (control, olaparib, IR, IR + olaparib). Treated xenografts were either irradiated by a single high-dose protocol of 25Gy (IR) at 24 hpi and/or treated with olaparib (renewed each 24 h for 4 consecutive days) (Figure 2A). After 4 days of treatment, the cytotoxic effects of the different treatment options were assessed by confocal microscopy.

Given the low proliferation rate of these tumors (mean~0.28% in VC8-B2^BRCA2wt^; mean~0.18% in VC8^BRCA2mut^), we could not detect any significant alteration in mitotic index in any of the conditions analyzed (Figure 2B–J).

Additionally, we analyzed nuclear size, since it can reflect chromatin modifications and changes in cell cycle dynamics, such as a G2 arrest as a consequence of DNA damage and repair [55,56,57]. Olaparib induced a significant increase in nuclear area size in VC8^BRCA2mut^ tumors (*** *p* < 0.001) but not in their BRCA2wt isogenic pair (*p* = 0.2853) (Figure 2K). Similarly, treatment with IR also induced an increase in nuclear size in VC8^BRCA2mut^ tumors (**** *p* < 0.0001), but not when the BRCA2wt gene was present.

As expected, and in accordance with previous in vitro studies [39], analysis of activated caspase 3 revealed that treatment with olaparib induced a ~2.2 fold increase in apoptosis in VC8^BRCA2mut^ (Figure 2L, **** *p* < 0.001), whereas in VC8-B2^BRCA2wt^ we only observed a ~1,5 fold increase (* *p* = 0.017; olaparib VC8-B2^BRCA2wt^ vs. olaparib VC8^BRCA2mut^ * *p* = 0.02). This higher increase in nuclear size and induction of apoptosis in *BRCA2* deficient tumors is in accordance with *BRCA1/2* mutations conferring a higher sensitivity to olaparib when compared to BRCA1/2 proficient tumors, as previously described in vitro [39,40].

The combination of IR with olaparib induced a significant increase in nuclear size in both BRCA2 mutant and wild-type tumors, as well as high levels of activated caspase 3, suggesting an additive effect of both therapies (Figure 2K–L). However, no differences in tumor size were detected in either cell line or treatment condition (Figure 2B–I,M). The induction of apoptosis has been described as a surrogate of treatment response preceding changes in tumor size—acting as an early monitor of tumor response—and thus allowing for shorter assays that can be easily included in the clinical decision-making time regarding treatment selection [58,59]. Thus, we anticipate that if the assay were to be extended, the tumor size effect would emerge, as we recently observed in other studies [60].

Altogether, our results show that BRCA2 mutations sensitize tumor cells to olaparib, and that combination of IR with olaparib may also constitute an option for BRCA2 proficient tumors. Moreover, we show that the zebrafish xenograft model can differentiate treatment responses in an isogenic model varying only in BRCA2 status. This highlights the potential of the zebrafish xenograft model for drug sensitivity profiling.

### 2.3. Human TNBC Zebrafish Xenografts Reveal Different Sensitivities to Olaparib and IR Independently of BRCA Status

To further validate our previous findings, we selected three different human triple negative breast cancer (TNBC) cell lines, representative of different BRCA and HRR status (Figure 3A,E,I and Appendix A)—Hs578T and MDA-MB-468 as BRCAwt and HCC1937 as BRCA1 mutant. We chose TNBC, since these are commonly associated with mutations in DDR pathways [61], making this molecular subtype a great candidate for therapeutics that exploit DDR deficiencies.

TNBC zebrafish xenografts were generated as previously described and randomly distributed amongst the different experimental conditions. At 5dpi (4 days of treatment), we assessed the impact of olaparib and its combination with IR on proliferation, nuclear size, apoptosis and tumor size (Figure 3).

Quantification of mitotic figures revealed no significant reduction in proliferation of any TNBC model upon treatment with olaparib (Figure 3A–M; Appendix A). In fact, we observed an increase in the percentage of mitotic figures in MDA-MB-468 tumors with this treatment (mean increase of ~42%, * *p* = 0.0459, Figure 3M). This was puzzling, since we were expecting a decrease in proliferation. However, recent reports suggest multiple roles for PARP in the structural machinery of mitosis, with PARP inhibition inducing DNA damage and leading to a possible metaphase arrest or mitotic abnormalities [62,63]. To further investigate this, we quantified the percentage of aberrant mitosis, displaying chromatin bridges, lagging chromosomes or asymmetric mitosis in MDA-MB-468 and HCC1937 tumors (Appendix A, Hs578T was not quantified due its very low mitotic rate**)**. Indeed, upon olaparib treatment, we could detect a significant increase in aberrant mitosis in MDA-MB-468 xenografts (* *p* = 0.0242), as previously reported in vitro [64], explaining the increase in mitotic counts. As for HCC1937 xenografts, after 4 days of treatment, only IR and the combined regimen led to an increase in mitotic aberrancies.

Next, we analyzed the impact of olaparib–PARPi treatment on induction of cell death by apoptosis and tumor size. As expected for BRCAwt tumors, and in accordance with previous studies [65,66], olaparib monotherapy did not induce cell death or reduction in tumor size in Hs578T xenografts (Figure 3B,O,P).

However, in the other BRCAwt model-MDA-MB-468, we detected a significant induction of activated caspase 3 upon olaparib (~47% increase, Figure 3O, *p* = 0.0167), illustrating how BRCAwt tumors can be sensitive to PARPi, in agreement with previous reports [9].

In BRCA1 mutant-HCC1937 xenografts, although we could not detect an induction of cell death by apoptosis, we could detect a reduction of ~24% (* *p* = 0.0488) of the tumor size upon olaparib monotherapy (Figure 3J,O,P). These results suggest that either the peak of apoptosis had passed, or that cells are dying by alternative mechanisms.

As previously described for the isogenic VC8/VC8-B2, we also analyzed the impact of IR and its combination with olaparib. In contrast to olaparib monotherapy, these treatments led to cytostatic effects, evident by the decrease in the mitotic index (Figure 3M), and to an increase in nuclear size in all three TNBC xenografts (Figure 3N, Appendix A). In the combinatorial regimen, we observed an increase in nuclear area of ~20% in Hs578T (*** *p* = 0.0003), ~45% in MDA-MB-468 (**** *p* < 0.0001), and ~30% in HCC1937 xenografts (**** *p* < 0.0001, Appendix A).

Most importantly, the combination of olaparib with IR induced apoptosis in all xenografts (~1.7 fold increase in Hs578T, **** *p* < 0.0001; ~4.6 fold increase in MDA-MB-468, **** *p* < 0.0001, and ~1.6 increase HCC1937, *** *p* = 0.0003, Figure 3C–D,G–H,K–L,O) and had a further synergistic effect on decreasing tumor size of BRCAwt Hs578T xenografts. These results are in agreement with previous in vitro studies [65].

In summary, we show that the zebrafish xenograft model can be used for olaparib sensitivity profiling, being able to identify BRCAwt resistant tumors (as expected-Hs578T), but also the unexpected BRCAwt sensitive tumors (MDA-MB-468). Furthermore, the effects observed with the combination of PARPi and IR show potential for a new treatment regimen to be used in a BRCA independent manner, where radiation may sensitize olaparib-resistant tumors.

### 2.4. Olaparib Can Inhibit Angiogenesis or Normalize Tumor Vessels

PARPs are pleiotropic molecules that play a role in different mechanisms associated with both pro- and anti-angiogenic effects [34]. Several reports have highlighted the role of PARP-1 in angiogenesis. Amongst other possible mechanisms, PARP-1 has been shown to not only promote hypoxia-inducible factor 2a (HIF-2a) transcription, but also bind directly to HIF-1a and HIF-2a, leading to their stabilization and activation. Accordingly, PARPi have been shown to lead to the reduction in tumoral angiogenic potential [67,68]. However, during ischemia, PARPi have also been reported to promote angiogenesis [69]. These contradictory reports highlight the pleiotropy of these molecules, and the possible divergent effects they may have in different individual contexts.

The same applies to IR. Depending on the dose and type of particles used, as well as the anatomical location to which it is applied, it can either inhibit or activate angiogenesis [70,71,72].

Given this duality of reports, and taking advantage of our functional model, we further investigated the effect of olaparib and its combination with IR on the tumor-related vessel network. To this end, xenografts were generated in *Tg(fli1:eGFP)* background and randomly distributed amongst the different treatment options.

First, we examined the isogenic VC8 and VC8-B2 pair. Quantification of vessel density at 5dpi revealed that olaparib significantly decreased angiogenesis in BRCA2 mutated tumors (Figure 4A–H, U, from ~22% in controls to ~14.9% in olaparib ** *p* = 0.0073). This effect was intensified by the combination with IR (dropping to ~10.2% vessel density *** *p* = 0.0001, olaparib vs. IR + olaparib, * *p* = 0.03). These results are in agreement with previously reported findings where PARPi can have a negative effect on tumor angiogenesis. Moreover, its effect seems associated with a lack of a functional BRCA2 protein.

We next analyzed the TNBC panel. Here, we could only detect an impact on angiogenesis in the BRCAwt Hs578T tumors, which have the highest angiogenic potential in the panel. Tumor vessel density decreased upon treatment with IR in monotherapy (from ~29% controls to ~23% with IR, * *p*= 0.0482) and was further decreased in the combinatorial regimen (to ~19% in IR + olaparib regimen, **** *p* < 0.0001) (Figure 4I–U’).

It has been shown that, in general, tumor-related vessels are poorly perfused and, consequently, not fully functional [73]. The anti-angiogenic rationale began with the goal of inhibiting angiogenesis in order to reduce the oxygen and nutrient supply of tumor cells. However, the current view is that, instead of just inhibiting angiogenesis, anti-angiogenic therapies can often normalize tumor vessels. This normalization may lead to an anti-tumoral effect by improving delivery of drugs and enhancing the impact of radiation [74].

Thus, to investigate the impact of olaparib on vessel functionality, we generated Hs578T xenografts in *Tg(fli1:eGFP; gata1:DsRed)* [75], in which erythrocytes are labelled with DsRed and the vascular system with eGFP (Figure 4V–X’). With this transgenic line, we were able to assess if the tumor-related vessels were functional, i.e., able to transport red blood cells. At 5dpi, Hs578T control xenografts were highly angiogenic, with a vast network of vessels surrounding and infiltrating the tumor. However, in the majority of the tumors, we could not find any erythrocytes inside the tumor-related vessels. In contrast, upon olaparib treatment, we could observe a clear increase in the number of tumors with red blood cells inside the vessels, suggesting that olaparib is able to increase vessel perfusion and normalize vessel function (from ~29% xenografts with functional vessels in control to ~55% with olaparib treatment, ** *p* = 0.0054, Figure 4Z).

Altogether, our data suggest that olaparib is able to modulate the tumor vessel network, either by inhibiting angiogenesis or by increasing tumor perfusion.

### 2.5. Combinatorial Treatment of Olaparib and IR Is Able to Reduce Metastatic Spread

PARP inhibition has been shown to have contradictory effects in the prevention or promotion of progression to advanced metastatic cancer through EMT [76,77]. Furthermore, IR has been shown to have pro- or anti-metastatic effects depending on the tumor type [78]. Given this duality, we investigated the impact of olaparib and IR treatment in the metastatic potential of TNBC zebrafish xenografts. We quantified the number of xenografts with micrometastasis in the caudal hematopoietic tissue (CHT) in the tail region, the most distant site from injection (Figure 5A). Our results show that the combination of olaparib and IR was able to reduce the incidence of CHT micrometastasis to half in MDA-MD-468^BRCAwt^ tumors (xenografts with micrometastasis: from ~39% in controls to ~21% in IR+ olaparib, * *p* = 0.0273) (Figure 5B). In contrast, in HCC1937 and Hs578T tumors, we could not detect any changes in the metastatic potential when challenged with either treatment option (Figure 5B).

Overall, our results suggest that the combined usage of olaparib and IR induces different cellular alterations that surpass cytotoxicity. Additionally, our data show the potential of the zebrafish xenograft model to assess the metastatic potential in response to olaparib and IR.

## 3. Discussion

PARPi arose as a renewed hope in the treatment of TNBC patients with mutations in BRCA1/2 genes. More recently, the discovery of a link between BRCA mutations and other types of cancer increased the relevance of this class of drugs beyond ovarian and breast cancer. However, not every patient with a BRCA mutation is sensitive to PARP inhibition, and there are patients with a BRCAwt status that show improvement with these drugs [79]. BRCA mutant tumors may have other genetic alterations that allow proficient HRR; whereas BRCAwt tumors may have mutations in the HRR pathway becoming sensitive to PARPi [80]. Therefore, the use of PARPi could be extended beyond patients with a BRCA mutation to patients who present HR deficiency, i.e., “BRCAness” or “HRR deficiency” phenotypes.

Unfortunately, due to the huge complexity and redundancy of many of the DDR-related pathways, no biomarker has been identified as a good predictor of PARPi sensitivity [12,14]. Approaches range from genetic and expression profiling of HR-related genes to promoter status or genomic scars. However, even in clinical trials that used ~133 different genetic biomarkers to allocate patients according to their HRR status, still some patients who were identified as HRR deficient did not benefit from olaparib [15,16].

This highlights the need for a different approach—looking at function and not just “markers” [16]. To this end, several assays have been developed to detect HR functional state scoring, with some promising results [28,29]. Nevertheless, PARPs are pleiotropic enzymes with functions that exceed the well-known DNA repair, and may impact in the tumor microenvironment, having a therapeutic value beyond cell-autonomous synthetic lethality [34].

Furthermore, even though PARPi are generally considered as a safer treatment option when compared to other alternatives, these drugs are associated with an increased risk of grade ≥3 side effects [79,81,82]. Therefore, there is need for a functional test that allows for the characterization of a tumor’s response to different treatment options, revealing not only cell intrinsic, but also TME-associated features in an assay independent of the genetic makeup/mechanisms involved.

Within this approach, treatment response to olaparib has been extensively studied in in vivo mouse models, through analysis of tumor size and survival rates [53,83,84,85,86,87]. Even though the mouse model is the gold standard, mouse patient derived xenografts take about ~2–4 months for sample engraftment and expansion, which makes it unfeasible for clinical decision-making [32].

The zebrafish model presents numerous advantages and has been used in several experimental approaches, including drug screenings and the generation of patient-derived zebrafish Avatars, using embryonic–larval stages or even adult zebrafish [30,31,33,88,89,90,91]. The optical transparency and single-cell resolution allow for the evaluation of different readouts apart from the classically used tumor size analysis [32,91].

Here, we tested the use of zebrafish larvae xenografts for a rapid characterization of response to olaparib PARPi, where we can detect not only the cell autonomous impact of treatments, but also its interplay with the tumor microenvironment and metastasis evaluation [30,31,33].

We show that we can distinguish BRCA2-dependent sensitivity to PARPi olaparib in isogenic VC8/VC8-B2 tumors, illustrating for the first time in this model the concept of synthetic lethality, in cells that only differ in their BRCA2 status.

Next, we tested human TNBC models with different BRCA status. As expected, in association with its BRCAwt status and its reported in vitro resistance to olaparib [65,66], Hs578T revealed no alterations in proliferation rate, no increase in activated caspase 3 or decrease in tumor size.

In contrast, the other BRCAwt model-MDA-MB-468 showed sensitivity to PARPi olaparib, i.e., we could detect an increase in aberrant mitotic figures and in activated caspase 3. These results are in agreement with previous in vitro [9,65,66] and in vivo studies (mouse xenografts) that show sensitivity of MDA-MB-468 to olaparib [83,84,85,87]. This unexpected sensitivity may be conferred by their known *PTEN* mutation, which has been shown to impair HRR [92,93].

Finally, in the BRCA1mut xenografts we could detect a significant decrease in tumor size. Although HCC1937 BRCA1mut cell line has been described to have a moderate sensitivity to PARPi olaparib in vitro [9,66,94] in vivo mouse xenografts show a reduction in tumor size upon olaparib treatment, similarly to our zebrafish model [95].

A lack of response to treatment with a PARPi is often associated with an acquired resistance after long term exposure [25,96]. Therefore, recent years have seen a surge of interest in the combination of PARPi with other therapies which may revert resistant phenotypes, expanding the indication of these drugs beyond HRR deficient tumors [96,97]. Currently, a number of ongoing clinical trials are assessing, amongst other schemes, the effects of the combination of olaparib and IR on a wide range of tumors, from breast and ovarian to glioblastoma and lung cancers [98,99,100,101,102]. This combination would entail a lower systemic toxicity than combining, for example, PARPi with chemotherapy. Since IR is already included in treatment guidelines in many different types of cancers, this association may soon be approved for locally advanced cancers treated with PARPi such as, for example, breast cancer.

In this sense, we investigated if IR could sensitize tumors to olaparib. Our results suggest that, indeed, this combination has potential benefits independent of BRCA1/2 status or olaparib basal sensitivity. This combinatorial treatment impacts tumorigenesis not just locally, increasing nuclear area and caspase 3 activation in all tested cell lines; but also systemically, differentially decreasing angiogenic and metastatic potentials in BRCAwt TNBC.

Finally, our work also shows the possibility of evaluating the non-cell autonomous effects of olaparib and IR on angiogenesis and metastasis formation, crucial hallmarks of cancer, difficult to assess in alternative models.

## 4. Materials and Methods

### 4.1. Animal Care and Handling

In vivo experiments were performed in the zebrafish model *(Danio rerio)*, were handled and maintained according to the standard protocols of the European Animal Welfare Legislation, Directive 2010/ 63/ EU (European Commission, 2016) and Champalimaud Fish Platform. The study protocol was approved by the Portuguese institutional organisations ORBEA (Órgão de Bem-Estar e Ética Animal/ Animal Welfare and Ethics Body) and DGAV (Direção Geral de Alimentação e Veterinária/ Directorate General for Food and Veterinary) 2015/005.

### 4.2. Zebrafish Lines

*Tg(fli1:eGFP)* has an eGFP under the fli1 promoter, expressed specifically in endothelial cells, allowing the visualization of both blood and lymphatic vascular systems. *Tg(fli1:eGFP;gata1:DsRed)* allows visualization of erythrocytes and their relative location to vessels. The transparent zebrafish line, casper, was also used for this project [103].

### 4.3. Cell Lines

VC8—originally derived from the V79 cell line—and VC8-B2—derived from the VC8 cell line, complemented with the human BRCA2 gene—were kindly donated by Thomas Helleday. Triple negative breast cancer cell lines Hs578T and MDA-MB-468, originally from American Type Culture Collection, were authenticated through short tandem repeat profiling karyotyping isoenzyme analysis. HCC1937 cell line was kindly provided by Kathleen Claes. All cell lines were tested for mycoplasma.

### 4.4. Cell Culture

VC8, VC8-B2, Hs578T and MDA-MB-468 cells were cultured in Dulbecco’s Modified Eagle Medium DMEM (Biowest, Nuaillé, France) supplemented with 10% fetal bovine serum FBS (Gibco, Thermo Fisher Scientific, Waltham, MA, USA) and 1% Penicillin−Streptomycin (Hyclone, Marlborough, MA, USA) in a humidified atmosphere containing 5% CO_2_ at 37 °C. For Hs578T, culture medium was supplemented with insulin at 10 μg/mL (Sigma-Aldrich, St. Louis, MO, USA). HCC1937 cells were cultured in RPMI (Biowest, Riverside, MO, USA) supplemented with 10% fetal bovine serum (Sigma Brazil, Cotia, São Paulo, Brazil) and 1% penicillin–streptomycin 10,000 U/mL (Hyclone, Marlborough, MA, USA) in the same atmospheric conditions.

### 4.5. Cell Labelling

Cell lines were labelled with Vybrant^TM^ CM-DiI (, Thermo Fisher Scientific, Waltham, MA, USA) at a concentration of 2 μL/mL or Deep Red (CellTrackerTM-Thermo Fisher Scientific, Carlsbad, CA, USA) at a concentration of 1 μL/mL according to manufacturer’s instructions. Cells were then resuspended to a final concentration of 0.50 × 106 cells/μL.

### 4.6. Zebrafish Xenografts Injection

Cancer cells were microinjected into the perivitelline space (PVS) of anesthetized 2dpf zebrafish. In general, ~800–1500 cells are injected, depending on their size (larger cells ~800; smaller cells ~1500). After injection, xenografts were transferred to 34 °C until the end of experiments. At 1 day post-injection (dpi), zebrafish xenografts were screened regarding the presence or absence of a tumoral mass. Xenografts with cells in the yolk sac, cell debris or non-injected zebrafish embryos were discarded, whereas successful ones were grouped according to tumor size.

### 4.7. Zebrafish Xenografts Irradiation and Drug Administration

After screening at 1 day post-injection (dpi), xenografts with similar tumor size were randomly distributed in the treatment groups: control E3 medium supplemented with dimethyl sulfoxide (DMSO), olaparib in E3, single high dose (SHD) of IR 25Gy followed by E3 medium supplemented with DMSO and SHD IR 25Gy followed by olaparib 50 µM in E3 for 4 consecutive days. We tested several concentrations of olaparib (see Appendix A) and chose 50 μM, ~10× the C_max_ (maximum plasma concentration found in patients) as our working concentration, since this was the concentration that showed tumor response in all cell lines without inducing mortality in the zebrafish embryo. Irradiation procedures and regimens were adapted for zebrafish xenografts by the Champalimaud Foundation Radiation Oncology Department as previously reported [30]. The 6MV X-rays beams with the corresponding prescription dose (Gy) were calculated with the same algorithm used in clinical practice (ECLIPSE, Varian Medical System, Palo Alto, CA, USA) and were delivered via a linear accelerator (Truebeam, Varian Medical Systems, Palo Alto, CA, USA). Irradiation was targeted to the center of a defined area of 30 × 30 cm where the 6-well plates with the anesthetized zebrafish were placed (3 mL of E3 medium per well). The plates were positioned with a source-to-surface distance of 100 cm.

### 4.8. Whole-Mount Immunofluorescence

5dpi zebrafish larvae were euthanized with tricaine methanesulfonate (MS-222 at a concentration of 4200 mg/L), fixed in 4% (v/v) formaldehyde (Thermo Fisher Scientific, Carlsbad, CA, USA) in phosphate-buffered saline (PBS) 0.1%Tween, overnight at 4 °C or for 2 h at room temperature and transferred to 100% methanol for long-term storage. Zebrafish xenografts were rehydrated by immersion in solutions with a decreasing concentration of methanol and washed and permeabilized with PBS–Triton 0.1% (w/v). After washing, zebrafish larvae were incubated for 7 min in ice cold acetone (−20 °C) for further permeabilization. Blocking for 1 h at room temperature was completed to prevent unspecific binding of the antibodies using PBDX-GS—for 50mL PBS 1X: 0.5 mL of DMSO, 250 μL of 10% Triton, 0.5 grams of bovine serum albumin (BSA) (Sigma-Aldrich) and 750 μL of 15 μL/mL goat serum (GS) (Sigma-Aldrich). Primary and secondary antibodies and DAPI were incubated for 1 h at room temperature followed by overnight incubation at 4 °C. Primary antibodies: anti-activated caspase 3 (rabbit, cell signaling, 1:100, reference: CST9661), anti-GFP (mouse, Roche, 1:100, reference: 11814460001) and anti-human HLA—MHC-class I (rabbit former MHC-class I, ab52922, ABCAM, Cambridge, UK,). Secondary antibodies: Alexa goat anti-rabbit 488 (Molecular probes, 1:400), anti-mouse 488 (Molecular probes, 1:400), and anti-mouse 647 (Molecular probes 1:400). Nuclei were counterstained with DAPI at 1:100. After staining, zebrafish xenografts were mounted in mowiol.

### 4.9. Imaging and Quantification

For data acquisition zebrafish xenografts were imaged in a Zeiss (Oberkochen, Germany) LSM710 fluorescence confocal microscope with an objective LD LCI Plan-Apochromat 25×/0.8 Imm Corr DIC M27 (Zeiss) with 5 μm interval z-stacks. Quantification analysis was performed using Fiji/ImageJ 1.8.0 software. For tumor size, Cell Counter Plugin was used and the number of total DAPI (tumor size) = mean (3 slices: Zfirst, Zmiddle, Zlast) × total no slices)/1.5. Tumor cells were identified by both DAPI pattern and CM-DiI labelling. Mitotic figures and activated caspase 3 were quantified manually in all slices of the stack and shown in percentage in relation to tumor size (total number of tumor cells). Data normalization was completed in relation to the control of each experiment to allow direct comparison between different cell lines.

### 4.10. Metastatic Potential Quantification

At 4dpi, metastatic potential was determined by quantifying the percentage of xenografts that present micrometastasis in the CHT in the population of xenografts analyzed for each condition.

### 4.11. Imaging and Angiogenesis Quantification

Vessel density was assessed throughout z-projections of corresponding images using the ImageJ Z Projection tool and the percentage of eGFP fluorescence per tumor was quantified. To analyze vessel infiltration, the superficial slices of the tumors were not considered. Vessel density = eGFP area (Fli1:eGFP labeled vessels)/tumor area. Vessel perfusion was determined through the percentage of xenografts with erythrocytes present inside the tumor vasculature in the population of xenografts analyzed for each condition.

### 4.12. Statistical Analysis

Statistical analysis was performed using GraphPad Prism software. Datasets were challenged by normality tests (D’Agostino and Pearson and the Shapiro–Wilk). Data with Gaussian distribution were analyzed by unpaired t-test. Datasets that did not pass the normality test were analyzed by the Man–Whitney test. All normalized data (fold induction or tumor size normalized) were analyzed by Mann–Whitney test. Differences were considered significant at *p* ≤ 0.05 and statistical output was represented by stars as non-significant (NS) > 0.05, * ≤ 0.05, ** ≤ 0.01, *** ≤ 0.001 and **** ≤ 0.0001. All graphs presented the results as mean ± standard error of the mean (SEM).

## 5. Conclusions

This work reveals not only the potential of the zebrafish xenograft model as a sensitivity profiling platform for olaparib, but also as a screening platform capable of determining the effects of different therapeutic combinations, in a time frame compatible with the clinical decision-making process. Future work aims at generating zebrafish patient-derived xenografts—zPDX/zAvatars—to further validate the model and test its predictive value in forecasting which patients will respond to olaparib therapy.

## Figures and Tables

**Figure 1 cancers-12-01769-f001:**
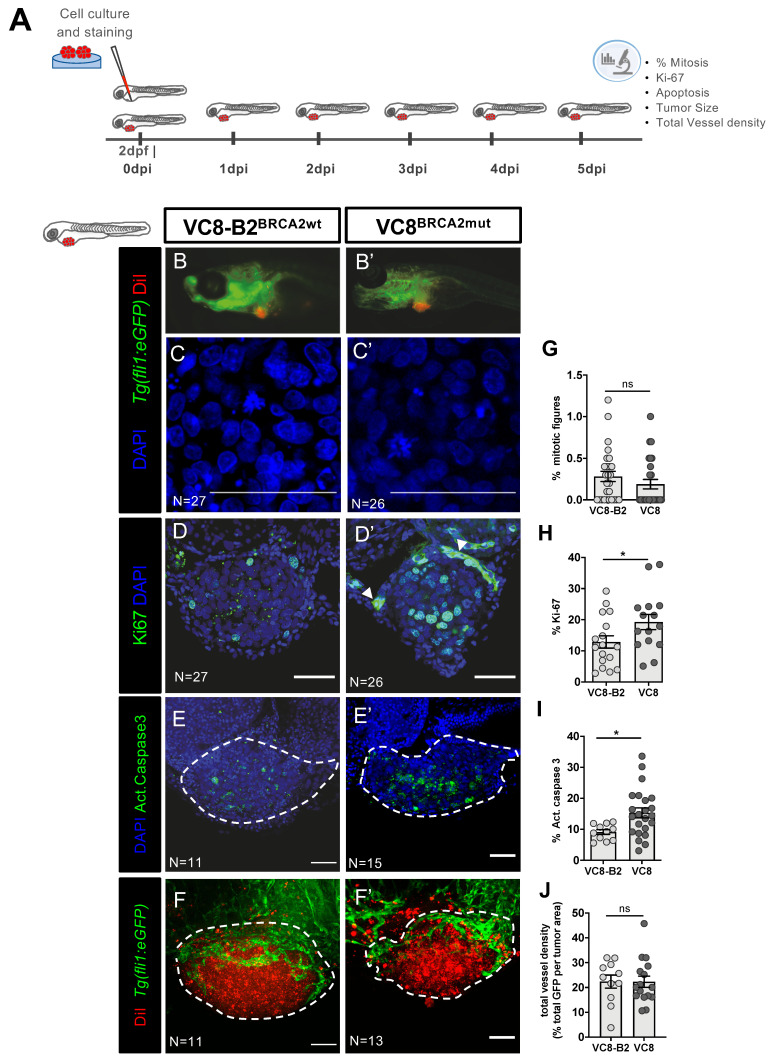
BRCA status can influence cancer-related hallmark phenotypes in a zebrafish xenograft model. VC8-B2^BRCA2wt^ and VC8^BRCA2mut^ hamster cell lines were fluorescently labeled with Vybrant CM-DiI (in red) and injected in the perivitelline space (PVS) of 2 days post fertilization (2dpf) *Tg(Fli1:eGFP)* zebrafish embryos (**A**). At 5dpi, zebrafish xenografts were imaged in the stereoscope (**B,B’**) and by confocal microscopy (**C**–**F’**). The percentage of mitotic figures (nuclei in blue) (**G**), Ki-67 (in green) (**H**), activated caspase 3 (in green) (**I**) and total vessel density (**J**, %GFP area—in green—relative to the whole tumor area) were quantified. All images are anterior to the left, posterior to right, dorsal up, and ventral down (as depicted in the scheme on top left). The dashed lines delineate the tumor area. Scale bar: 50 µm. Results are from three independent experiments and expressed as mean ± SEM, each dot represents one xenograft. The total number of xenografts analyzed is indicated in the images. Statistical results: not significant (ns) > 0.05, * *p* ≤ 0.05.

**Figure 2 cancers-12-01769-f002:**
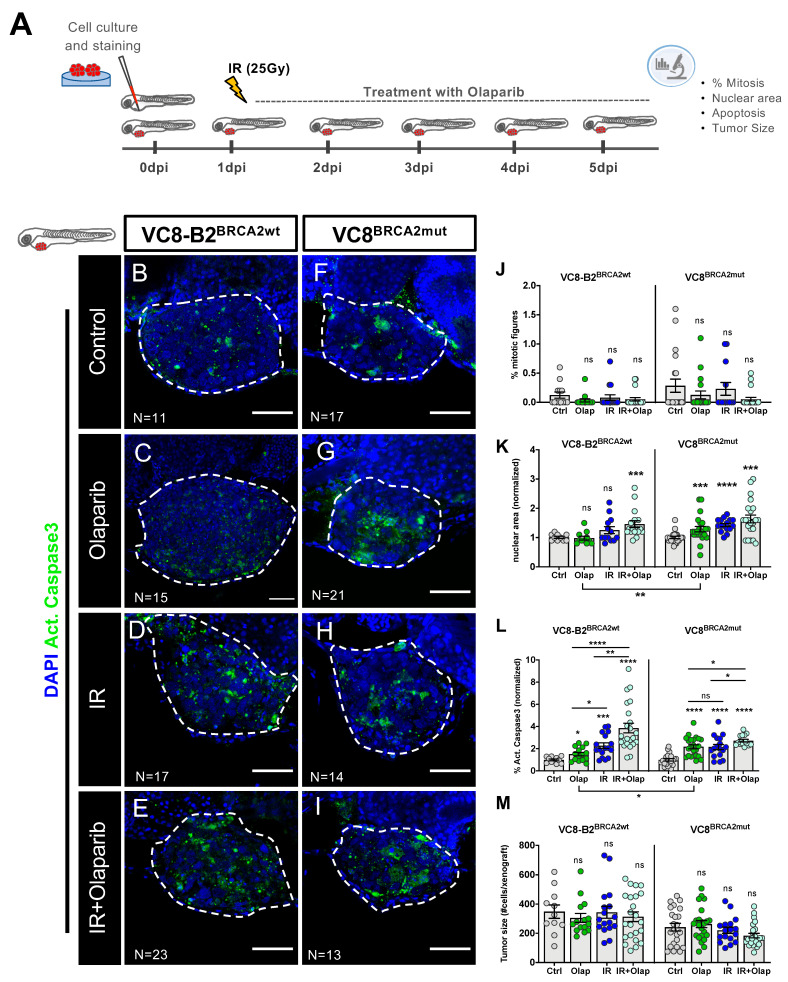
VC8 and VC8-B2 zebrafish xenografts show different sensitivities to treatment with olaparib and IR. VC8-B2^BRCA2wt^ and VC8^BRCA2mut^ cell lines were fluorescently labeled with CM-DiI (not shown) and injected into the perivitelline space (PVS) of 2dpf zebrafish embryos. At 24 h post-injection (24 hpi), zebrafish xenografts were screened and randomly distributed into the different experimental conditions: control, olaparib, IR, IR + olaparib. Xenografts were treated for 4 consecutive days and fixed at 5dpi (**A**). Zebrafish xenografts were sacrificed and fixed at 5dpi and prepared for confocal microscopy by immunolabeling for activated caspase 3 (marker for cellular death, in green) and nuclei staining with DAPI (in blue) (**B**–**I**). Mitotic index (**J**), nuclear area size (**K**), cell death-activated caspase 3 (**L**) and average tumor size (number of human DAPI cells) (**M**) were analyzed by confocal microscopy and quantified. The % of activated caspase 3 cells was normalized to respective controls to compare between different xenografts in different conditions. All images are anterior to the left, posterior to right, dorsal up, and ventral down (as depicted in the scheme on top left). The dashed line delineates the tumor area. Scale bar: 50 µm. Results are from three independent experiments and expressed as mean ± standard error of the mean (SEM), each dot represents one xenograft. The total number of xenografts analyzed is indicated in the images. Statistical results: ns > 0.05, * *p* ≤ 0.05, ** *p* ≤ 0.01, *** *p* ≤ 0.001, **** *p* ≤ 0.0001.

**Figure 3 cancers-12-01769-f003:**
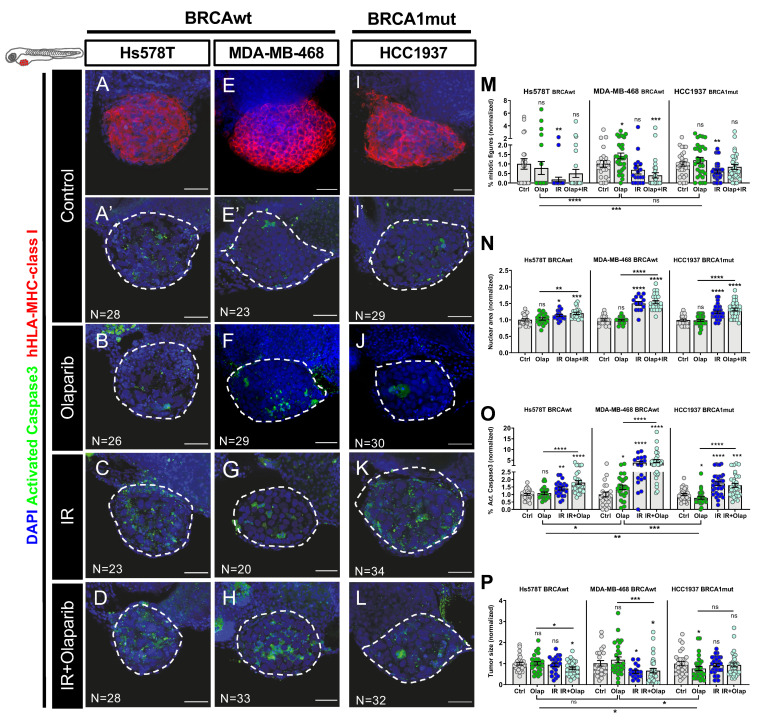
Zebrafish xenografts of triple negative breast cancer (TNBC) reveal different responses to olaparib and ionizing radiation (IR) independently of BRCA status. TNBC cell lines (Hs578T^BRCAwt^, MDA-MB-469^BRCAwt^ and Hcc1937^BRCA1mut^) were fluorescently labeled with CM-DiI (not shown) and injected into the perivitelline space (PVS) of 2dpf zebrafish embryos. At 24 hpi, zebrafish xenografts were screened and randomly distributed into the different experimental conditions: control, olaparib, IR, IR + olaparib. Xenografts were treated for 4 consecutive days and fixed at 5dpi. Representative confocal images of xenografts labelled with anti-human-leukocyte antigen-major histocompatibility complex (HLA-MHC)-class I (in red) (**A,E,I**) and anti-activated caspase 3 (in green) (**A’**–**L**). Nuclei were stained with DAPI (in blue) (**A**–**L**). Mitotic index (**M)**, nuclear area size (**N**), cell death-activated caspase3 (**O**) and average tumor size (number of human DAPI cells) (**P**) were analyzed by confocal microscopy and quantified. The % of activated caspase 3 cells and tumor size were normalized to respective controls to compare between different xenografts in different conditions. All images are anterior to the left, posterior to right, dorsal up, and ventral down (as depicted in the scheme on top left). The dashed line delineates the tumor area. Scale bar: 50 µm. Results are from three independent experiments and expressed as mean ± SEM, each dot represents one xenograft. The total number of xenografts analyzed is indicated in the images. Statistical results: ns > 0.05, * *p* ≤ 0.05, ** *p* ≤ 0.01, *** *p* ≤ 0.001, **** *p* ≤ 0.0001.

**Figure 4 cancers-12-01769-f004:**
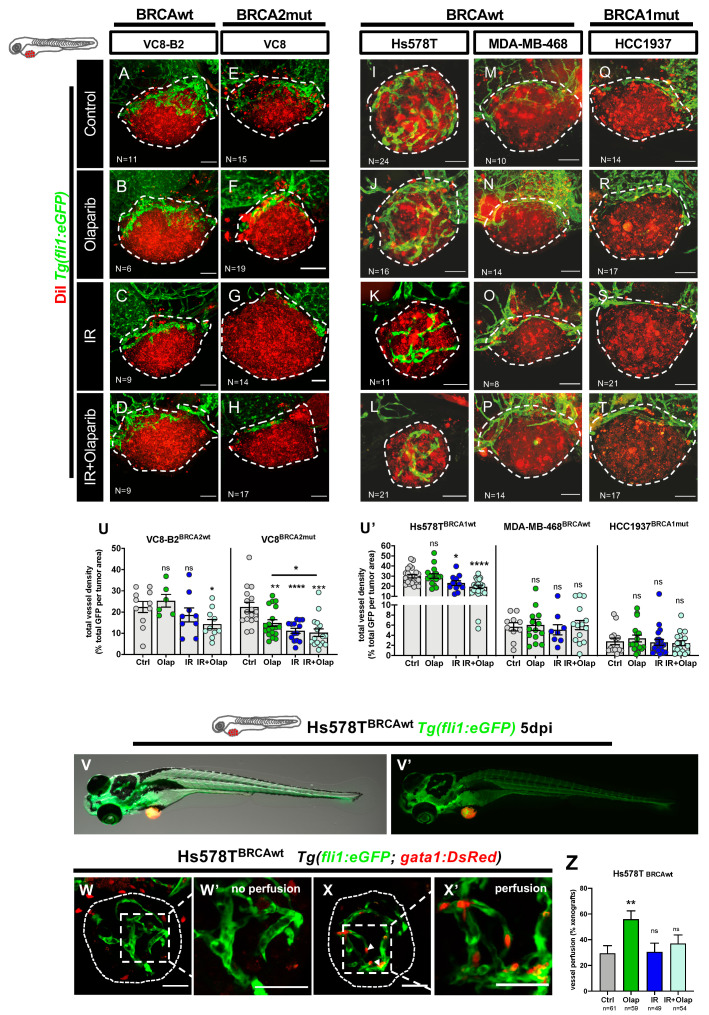
Olaparib and IR can have modulating effects on the tumor vascular network. All cell lines were fluorescently labeled with CM-DiI (in red) and injected in the perivitelline space (PVS) of 2dpf *Tg(fli1:eGFP)* zebrafish embryos. At 24 hpi, zebrafish xenografts were screened and randomly distributed into the different experimental conditions: control, olaparib, IR, IR + olaparib. Xenografts were treated for 4 consecutive days, fixed at 5dpi and imaged by confocal microscopy for analysis of vessel density (**A**–**T**). The quantification of vessel density is represented by percentage of tumor area occupied by vessels (in green) (**U**). To analyze vessel functionality, TNBC cell line Hs578T was fluorescently labeled with CellTracker™ Deep Red Dye (in red-false color) and injected in the PVS of 2dpf *Tg(gata1:RFP;fli1:eGFP).* Hs578T xenografts were screened and randomly distributed in the different experimental conditions: control, olaparib, IR and IR + olaparib. Representative images of 5dpi Hs578T xenografts (**V**,**V’**). The presence of erythrocytes (in red) inside the tumor-associated vessels (in green) was scored as: absence of erythrocytes = no perfusion (**W**,**W’**) or presence of erythrocytes = with perfusion (**X**,**X’**) and quantified (**Z**). All images are anterior to the left, posterior to right, dorsal up, and ventral down (as depicted in the scheme on top left). The dashed lines delineate the tumor area. White arrowheads indicate erythrocytes inside the vasculature. Scale bar: 50 µm. Results are from three independent experiments and expressed as mean ± SEM, each dot represents one xenograft. The total number of xenografts analyzed is indicated in the images. Statistical results: ns > 0.05, * *p* ≤ 0.05, ** *p* ≤ 0.01, *** *p* ≤ 0.001, **** *p* ≤ 0.0001.

**Figure 5 cancers-12-01769-f005:**
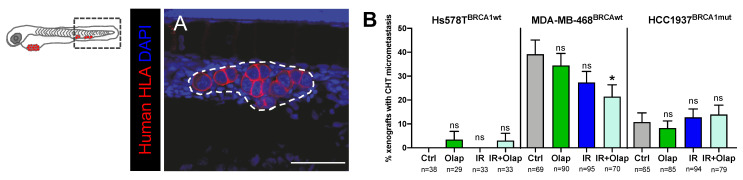
Metastatic potential of TNBC cell lines is modulated by olaparib and IR. TNBC (Hs578T^BRCAwt^, MDA-MB-469^BRCAwt^ and Hcc1937^BRCA1mut^) xenografts were generated as previously described and at 24 hpi, were screened and randomly distributed into the different experimental conditions: control, olaparib, IR, IR + olaparib. At 5dpi and 4 days post-treatment, xenografts were fixed to analyze the presence of micrometastasis in the caudal hematopoietic tissue (CHT) (**A**). Representative image of a micrometastasis in the CHT is labelled with anti-human-HLA–MHC-class (in red) for human cell identification. The dashed white line represents the micrometastasis area, scale bar: 50 µm (**A**). The metastatic potential was quantified as the percentage of xenografts that present micrometastasis in the CHT at 5dpi (**B**). Results are from three independent experiments and expressed as mean ± SEM. The total number of xenografts analyzed is indicated below the chart. Statistical results: ns > 0.05, * *p* ≤ 0.05.

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
