# Peer review of "Zebrafish Xenografts Unveil Sensitivity to Olaparib beyond BRCA Status"

_cancers, 2020, doi:10.3390/cancers12071769_

Round 1

Reviewer 1 Report

The author present a sound and important evaluation of the zebrafish platform. Nevertheless, there are some major points that have to be clarified prior to publication.

Major comments:

  • Conclusion of the abstract seems to be too ambitious, did they show feasibility for a co-clinical trial by using commercial cell lines?
  • General: No references in the results part. The interpretation should be part of the discussion not the results. In contrast, no reference to other related work of other groups could eb found in the discussion. The references to other zebrafish platforms is completely missing. A discussion about the different read-outs and their pro-s and cons is missing
  • As this is a rather new assay a positive control would be helpful, similar to Staurosporin in an in vitro assay. Thereby, the different read-outs and the sometimes confusing outcomes would be easier to interpret.
  • A dose -response curve for olaparib as well as IR is highly recommended to better validate the usefulness of the different read-outs.

Minor comments:

Figure 1: although described very clear, I do not understand the scheme on the top of the figure. Maybe it is not depicted completely?

Some typos and grammatical mistakes have to be eliminated

Author Response

We thank Reviewer #1 for carefully reading our manuscript and raising important points, which we will try to address point by point. Please see attachment.

Reviewer 2 Report

In the work presented in this manuscript, Ana et al. explore the utility of cell-line xenografts in the zebrafish for studying anti-tumor efficacy of the PARP inhibitor Olaparib and show that this model can be used to identify cancer that is olaparib-sensitive in spite of being normal for BRCA. The authors also show that the model can be used to test efficacy of combining olaparib with radiation therapy. Because of ease of visualization, zebrafish is a good experiment model to study cells in the context of the full organism with multiple tissues. The authors have taken advantage of this to examine efficacies of anti-cancer regimens against human cancer cells. Not only the study's findings, such as that cells with normal BRCA can be sensitive to PARP inhibition, but also the model used in the work are interesting.

Comments.

Result 2.1: The xenografts of BRCA2-wildtype and -mutant VC8 cell-lines in zebrafish were different for only Ki-67 and Caspase 3 activity but not cell proliferation, angiogenesis, or metastasis. It is therefore difficult to conclude that “these results show that we can detect differential tumor behaviors of these BRCA2 isogenic cell lines in the zebrafish model..” How do these two cell-lines differ for the latter features in other systems (such as xenograft in another animal model, or in vitro cell culture)? And how much do they differ in terms of DNA repair activity? Please add this information in Discussion.

Result 2.3: A reduction in tumor size is normally essential for assessing the potential of new drug treatment. Without a change in tumor size, how much can truly be concluded about the efficacy of drug/IR treatments? Please discuss this point in Discussion.

Result 2.4: Some of the sample sizes appear to be incorrect. For example, although N is 1 in figure H, the sample number of IR+Olap group in U is more.   Also, could functionality of tumor-related vessels have been assessed for VC8-B2 or VC8?

Result 2.5: The authors concluded that “the combined usage of olaparib and IR induces different cellular alterations that surpass cytotoxicity, modulating the tumor microenvironment.” However only different therapeutic effects are shown here depending on the tumor cell type, and it is somewhat exaggerating to conclude this way.

There is no description of **** in legend.

How are tumors growing from xenografted cell-lines identified in the zebrafish, e.g., to outline them in the various figures? This should be mentioned in Methods.

Please provide corresponding light microscopy images for some of the fluorescent microscopy images, perhaps in Supplementary Material. Readers will also like to see wide-view images of some fishers with the tumors.

Although a reference has been provided, some detail about the irradiation method should be presented (name of irradiator instrument, etc.).

The use of “AVG” for “average” is unusual for scholarly publications. I suggest using “mean”, which is understandable by all.

Some abbreviations in figure legends are not explained. E.g., “PVS” in Fig. 1. Explanations should be put in the figure legends for all abbreviations, except commonly used ones like “SEM”. In general, figures and their legends should be fully understandable by themselves without a need to look at the rest of the paper.

Comma (,) is used to indicate the decimal point at many places. Period (.) should be used instead.

Mistake in Methods section – Zebrafish Xenografts Preservation – “5dpi, xenografts were euthanized…”: the fish were euthanized.

After all, it is nuclear which model is judged to be effective for these treatments. Since the tumor size has not decreased, it is insufficient as a preclinical model for patient selection.

Author Response

We thank Reviewer#2 for carefully reading our manuscript and raising important points, which we will try to address point by point. Please see attachment.

Reviewer 3 Report

The paper of Varanda et al proposes the use of zebrafish xenotransplants to predict sensitivity for olaparib.

  • General remark on statistics: authors specify the numerosity but not the number of independent assays.
  • General remark on the staining: DiI is not sufficient to discriminate xenotransplanted cells or to distinguish them from resident immune cells phagocyting cell debris. A species-specific Ab needs to be used which distinguishes between the cell of the host and the xenotransplanted cells
  • General remark on the biomarker: the percentage of proliferating cells is too low, mitotic figures is not a good marker for any discussion/interpretation
  • General remark on models: the cell lines used (Hs578T, MDA-MB-468, HCC1937) are not good for the purpose of this study. Response to olaparib is controversial. Indeed, the study is not well posed as the aim of the study is to validate the use of zebrafish xenograft model as a sensitivity profiling platform for olaparib
  • A small increase in ki-67 and caspase positive cells is not sufficient to state “we can detect differential tumor behaviors of these BRCA2 148 isogenic cell lines in the zebrafish model”. This sounds as an overstatement
  • The authors selected ~10 times the Cmax (50mM) as working concentration. This is totally arbitrary. I suggest to perform HPLC analysis to correlate drug concentration in fish and humans
  • The higher sensitivity to olaparib in BRCA2 deficient tumors when compared to BRCA1/2 190 is very marginal and not confirmed in the combined treatment (olaparib + IR). I guess that the variability of the xenotransplant and cell engraftment influence data. For example, I am not convinced that panel C and G in figure 2 are comparable situations because the staining pattern is totally different.
  • The authors show that human TNBC zebrafish xenografts reveal different sensitivities to olaparib and IR independently of BRCA status. Has this been confirmed by other models? When validating a new model, a comparison with another well accepted model is necessary otherwise it would impossible to get into any conclusion. Is sensitivity to olaparib and IR independently of BRCA status or is the zebrafish model not properly working?
  • Figure 4: in some panels the number is 1 or 2, in O is not specified. This is not acceptable. In some panels I could detect SIV, it is not clear how authors distinguish angiogenesis from tumour neo-angiogenesis
  • In figure 4, U’: I see a clear different response among the different cell lines but, within a cell type, differences, with respect to the treatment, are marginal
  • Figure 5B (same consideration than before): I see a clear different response among the different cell lines and different response to the treatments is not comparable

Minor

Move table 1 in supplementary or M&M

Line 53 “BER complex” not defined

Line 100 “TME” not defined

Line 123 “2 days post fertilization (2dpf) zebrafish larvae”, at the best of my knowledge the arbitrary end of the zebrafish embryonic period is ~72 hpf at 28.5 °C (Kimmel et al. 1995). I suggest to replace “larvae” with “embryos”.

Line 130 please replace Ki67 with Ki-67

and in general pay attention to be coherent in all the manuscript

Figure 1 C please check the arrowhead

Line 157 “as depicted in the scheme on top left” problem with the image, I can see only the eye as a gray spot and the red mass

Line 182 Please be careful to use dot-decimal notation. Check all the manuscript.

Line 431 please add the incubation time

Line 454 please add time and temperature of fixation

Line 455 “We designed the test to span 5 days not only due to animal ethics constraints..” please be careful because 5 days in the Legislation on zebrafish is referred to the age. Your method finishs at 7 days post fertilization when the zebrafish are subjected to legislation governing animal testing.

Line 460 please provide the method or the reference for the Whole-mount immunofluorescence protocol performed.

Line 467 please add the image magnification that you use for the Quantification analysis.

Author Response

We thank Reviewer#3 for carefully reading our manuscript and raising important points, which we will try to address point by point. Please see attachment.

Reviewer 4 Report

The manuscript entitled “Zebrafish xenografts unveil sensitivity to olaparib beyond BRCA status” by Varanda et al., describes the use of the zebrafish xenograft model to assay for tumor cell response to PARP inhibitor olarparib alone or in combination with ionizing radiation. The effect of the therapies is measured according to 4 hallmarks of cancer: cell proliferation, cell death, angioginic and metastatic potential.

The manuscript is clear and very well presented and written. The experiments are well conducted and the conclusions are supported by the experimental data. Although based on a limited number of cell lines, the results revealed an olaparib sensitivity independent from the BRCA2 mutation status and showed that olaparib normalizes tumor vessels in double transgenic recipient zebrafish. This work illuminate the potential of the zebrafish xenograft model to predict how patients will respond to olaparib/IR therapy in a personalized medicine context.

As an improvement to the manuscript, I would suggest the authors to discuss in more details the choice of their therapeutic treatments in comparison to what is currently applied in clinics.

Author Response

We thank Reviewer#4 for carefully reading our manuscript and raising an important point, improving greatly our manuscript.

"As an improvement to the manuscript, I would suggest the authors to discuss in more details the choice of their therapeutic treatments in comparison to what is currently applied in clinics."

We thank Reviewer#4 for the suggestion and have now discussed in more detail our rationale (see lines 470-478).

Round 2

Reviewer 1 Report

the authors have addressed all major and minor comments and significantly improved the manuscript which is now ready for publication from my perspective

Reviewer 2 Report

I appreciate the time and effort the team has taken to address all the concerns, with new figure and more detailed methods that support the results and conclusions.